# Implications of Hyperoxia over the Tumor Microenvironment: An Overview Highlighting the Importance of the Immune System

**DOI:** 10.3390/cancers14112740

**Published:** 2022-05-31

**Authors:** Ana Belén Herrera-Campos, Esteban Zamudio-Martinez, Daniel Delgado-Bellido, Mónica Fernández-Cortés, Luis M. Montuenga, F. Javier Oliver, Angel Garcia-Diaz

**Affiliations:** 1Instituto de Parasitología y Biomedicina López Neyra, CSIC, 18016 Granada, Spain; anabelenherrera10@ipb.csic.es (A.B.H.-C.); estebanzm95@ipb.csic.es (E.Z.-M.); ddelgado-ibis@us.es (D.D.-B.); monica.fernandez@ipb.csic.es (M.F.-C.); 2Consorcio de Investigación Biomédica en Red de Cáncer (CIBERONC), 28029 Madrid, Spain; lmontuenga@unav.es; 3Program in Solid Tumors, CIMA-University of Navarra, 31008 Pamplona, Spain; 4Navarra Health Research Institute (IDISNA), 31008 Pamplona, Spain

**Keywords:** tumor microenvironment, hypoxia, hyperoxia, immunotherapy, inflammation

## Abstract

**Simple Summary:**

The local conditions of tumor cell growth, known as the tumor microenvironment (TME), are characterized by low oxygen supply (hypoxia) caused by insufficient blood delivery. Hypoxic cancers have a strong invasive potential, metastasis, resistance to therapy, and a poor clinical prognosis. The use of supplemental oxygen, known as hyperoxia, has been described to diminish the hypoxic state and to achieve a better response to different treatments. Here, we provide an overview of how hyperoxia interacts with other therapies decreasing tumor progression and the negative effects of the use of high oxygen levels. We also perform an analysis, showing the differences in the patterns of expression between a tumor-derived cell line and a nonmalignant cell line.

**Abstract:**

Hyperoxia is used in order to counteract hypoxia effects in the TME (tumor microenvironment), which are described to boost the malignant tumor phenotype and poor prognosis. The reduction of tumor hypoxic state through the formation of a non-aberrant vasculature or an increase in the toxicity of the therapeutic agent improves the efficacy of therapies such as chemotherapy. Radiotherapy efficacy has also improved, where apoptotic mechanisms seem to be implicated. Moreover, hyperoxia increases the antitumor immunity through diverse pathways, leading to an immunopermissive TME. Although hyperoxia is an approved treatment for preventing and treating hypoxemia, it has harmful side-effects. Prolonged exposure to high oxygen levels may cause acute lung injury, characterized by an exacerbated immune response, and the destruction of the alveolar–capillary barrier. Furthermore, under this situation, the high concentration of ROS may cause toxicity that will lead not only to cell death but also to an increase in chemoattractant and proinflammatory cytokine secretion. This would end in a lung leukocyte recruitment and, therefore, lung damage. Moreover, unregulated inflammation causes different consequences promoting tumor development and metastasis. This process is known as protumor inflammation, where different cell types and molecules are implicated; for instance, IL-1β has been described as a key cytokine. Although current results show benefits over cancer therapies using hyperoxia, further studies need to be conducted, not only to improve tumor regression, but also to prevent its collateral damage.

## 1. Introduction

Oxygen homeostasis is indispensable to maintain an appropriate biological functioning. For this purpose, there are chemosensory systems which ensure the proper oxygen supply to the cells. A reduction in oxygen concentration is a phenomenon known as hypoxia [1]. The hypoxia-inducible factors (HIFs) are the main regulators of this state, not only proving its importance in physiological processes that occur under low oxygen levels such as placental or heart development during embryonic development, but also linking to pathologies such as cerebral and myocardial ischemia or cancer [1,2,3]. The importance of reverting this hypoxic state in tumor progression through the use of supplemental oxygen has increased in recent years and is described in this review.

Hypoxia is known to play an important role in cancer progression inducing cellular responses which are associated with the adaptation to nutrient deprivation, cell proliferation, tumor invasion, and metastasis. Next-generation sequencing and microarray techniques have allowed understanding the molecular mechanisms of this response through the identification of hypoxia-related genes [4,5]. Specifically, the role of HIF-1 in the regulation of genes that control angiogenesis has been described, mainly through the overexpression of the vascular endothelial growth factor (VEGF) [6,7,8], glycolytic metabolism [9,10,11,12,13], tumor proliferation, or the immune microenvironment [11,14]. The transcription factor HIF is a heterodimeric factor consisting of an oxygen-regulated alpha subunit and a constitutively expressed beta subunit. Three isoforms of the alpha subunit have been described (HIF-1α, HIF-2α, and HIF-3α). The oxygen value considered normal varies depending on the tissue. The term “physoxia” defines this physiological oxygen concentration, which is different from the atmospheric O_2_ concentration (21%) due to the transport that occurs when the oxygen is taken until it reaches the final destination [15,16,17,18,19]. When oxygen availability is lower than the physoxic range, the alpha subunit is stable and dimerizes with the beta subunit, the heterodimer is able to bind the transcriptional coactivator CREB-binding protein (CBP)/p300, and the complex recognizes the hypoxia response element (HRE) sequence of the target genes activating the transcription [20,21,22]. Therefore, each tissue presents different oxygen concentration that differs between normal and cancer tissue, as presented in Figure 1 [19].

Tumor structure was described by Thomlinson and Gray in human lung cancer, who proposed the presence of an oxygen gradient which leads to a necrotic area where oxygen cannot diffuse, surrounded by stroma live cells that obtain oxygen and nutrients from capillary vessels, while the tumor area is characterized by hypoxic cells immersed between those two areas [23,24]. Tumor hypoxia is an important barrier to overcome when successfully implementing cancer therapy. One approach would be through the use of supplemental oxygen to counteract this aggressive characteristic of the tumor microenvironment (TME); this technique has been previously used in other pathologies such as traumatic brain injury, ischemic stroke, or neurological conditions [25]. The present review discusses different aspects in which hyperoxia seems to be involved, highlighting its use as a promising novel cancer therapy. We also present a complete RNA-Seq transcriptional profile analysis of normal pulmonary epithelium cells, compared to metastatic melanoma cells under hyperoxic conditions. This study reveals the main cellular functions and genes affected by hyperoxia in normal and tumor cells that can lead to novel therapies and research approaches.

## 2. Impact of Hyperoxia on Different Cancer Therapies

### 2.1. Conventional Therapies and Hyperoxia

Solid tumors are characterized by the absence of a regular vasculature; in contrast, they present a vasculature that is continuously growing to satisfy the needs of rapidly proliferating cells, which could result in an aberrant network [26]. One of the factors contributing to resistance to cancer therapies has been attributed to low oxygen levels. The lack of a functional vasculature blocks the drug supplying from the blood vessels to the tumor cells, resulting in an ineffective action of chemotherapeutic agents. Furthermore, radiotherapy is focused on tumor cell DNA damage to achieve cell death produced by a high quantity of oxygen reactive species (ROS) that are produced after the ionizing radiation of the tissue; therefore, effectiveness is reduced under hypoxic conditions [27,28]. An increase in oxygen levels would potentiate not only the ROS levels inducing tumor cell death but also a more efficient drug delivery to the tumor [29]. Here, we present a variety of ways in which hyperoxia can affect different biological processes and their relationship with the most commonly used antitumor treatments.

#### 2.1.1. Chemotherapy

Hyperbaric oxygen therapy (HBO), which is based on the inhalation of elevated oxygen concentration at high pressure, is highly effective due to its capacity to overcome HIF-1α expression, thereby downregulating its target genes [30]. Solid tumors are characterized by the presence of a dense collagen network and elevated interstitial fluid pressure that hinder the transport of the agents to the cells. These two parameters are reduced after single and repeated HBO treatment, but an increase in the uptake of chemotherapeutic drugs by mammary tumors of rats has been observed after only a single treatment. As repeated HBO treatment has an antiangiogenic effect, reducing the mean vessel density and increasing diameter [31], it has been proposed that this will lead to a normalized vasculature with lower permeability [32]. After 3 days of therapy in a liver cancer model, a reduction in the collagen deposition was observed, as well as a better penetration and accumulation of a chemotherapeutic nanomedicine approved by the Food and Drug Administration (FDA), Doxil. One of the ways that hypoxia makes tumors resistant to chemotherapy is through cell-cycle arrest in the G0/G1 phase, but it is during DNA replication when this drug conveys its action. The supplemental oxygenation results in cells combating the arrest and sensitizing them to the agent, with the combinatory therapy having a synergistic antitumor effect [29,30]. An increase in chemotherapeutic response when HBO is used, through the decrease in tumor growth, has also been described in a prostate cancer model [33]. Combined HBO and chemotherapy have been performed in clinical trials; specifically, it was reported that 13 out of 15 patients with breast cancer treated with HBO and chemotherapy achieved tumor regression compared with just four out of 15 treated only with chemotherapy [34]. HBO and these drugs are currently applied in several clinical settings, and these studies also demonstrate that the combination carries no risk [30,34]. 

As opposed to HBO, normobaric oxygen therapy (NBO) is high oxygen administration at normal pressure [35]. Despite both therapies diminishing hypoxic regions, at high pressures, it is possible to reach better oxygenation of the tumor [36,37]. It was reported by Kim et al. colleagues that NBO inhibited the growth of lung cancer, which may have been due to an increase in ROS levels and apoptotic pathways [38]. In addition, the treatment of glioblastoma multiforme (GBM) cells with the chemotherapeutic agent temozolomide (TMZ) and normobaric hyperoxia shown that hyperoxia increased the cytotoxicity of the drug, reflected in a high cell death rate, which seemed to be caspase-dependent apoptosis as an increase in caspase 3 activity was observed, along with an increase in Bax/Bcl-2 ratio. Furthermore, this attenuation of TMZ resistance is not only enhanced by the apoptotic pathway, but has also been observed to activate the MAPK pathway [25]. This increase in the expression of caspase 3 and the Bax/Bcl-2 ratio was recently confirmed in lung tissues of mice treated with carboplatin chemotherapy and hyperoxia. Moreover, there were a higher number of TUNEL-positive apoptotic cells compared with each treatment alone, suggesting that they arose from a significantly increase in the oxidative stress. This combinatory therapy has a tumoricidal effect reflected in lung tumor volume and number decrease with a recovery in the structure [39]. 

The focus on the design of nanoparticles to modify the TME has increased over the past years. The structures of nanocarriers capable of improve the transport of chemotherapeutic agents, as well as release O_2_ selectively in cancer cells, have decreased the hypoxic state with a downregulation of HIF-1α expression, resulting in a reduction in tumor growth and metastasis. Moreover, these designs are considered biosafe due to their ability to affect tumor cells but not normal cells [40,41]. 

#### 2.1.2. Radiotherapy

To overcome the difficulties of cancer radiotherapy, the use of oxygen as an adjuvant therapy has also been proposed. The use of 100% NBO delayed tumor growth and reached local tumor control, which means that there was an increase in the radiosensitivity in a mouse mammary carcinoma [42]. As CO_2_ is a vasodilator, some authors have used it with the objective of increasing blood flow. Patients with head-and-neck tumors under hyperoxic treatment (98% O_2_ + 2% CO_2_) achieved a better tumor oxygenation that could improve the response to ARCON (accelerated radiotherapy with carbogen and nicotinamide) [43]. HIF-1α has also been previously described to participate in the induction of an antiapoptotic program through the Akt/ERK pathways [44]. Human HELA cells placed under hyperoxia conditions (95% O_2_ + 5% CO_2_) decreased the activation of these two proteins, which increased their radiosensitivity and apoptosis, as well as downregulated HIF-1α and its target gene, VEGF [45]. Radiation therapy is also used as a treatment for central nervous system tumors; although it is effective and safe, it can produce radiation necrosis. To reduce this complication, patients have been treated with HBO, resulting in a clinical improvement in 92% of cases. This study shows how supplemental oxygen can reduce the side-effects of conventional therapies, although further evaluations have to be performed [46].

Triple-negative breast cancer (TNBC) is well known to be a very aggressive tumor, but it was not until a few years ago that Mast et al. [47] described it as a highly hypoxic kind of tumor, occurring at the early stages of the cancer development. They demonstrated that TNBC xenograft tumors under daily 100% oxygen administration were able to inhibit tumor growth. A chemotherapeutic agent, paclitaxel (Pax), was administered in combination with oxygen, increasing tumor pO_2_ and inhibiting tumor growth. In addition, when radiation therapy was applied to those treated with the combination, the result was a significant decrease in tumor growth [47].

### 2.2. Cancer Immunotherapy and Hyperoxia

As shown in Figure 2, the use of supplemental oxygen has demonstrated the ability to interfere with several immunological pathways, motivating great interest for the implementation of current immunotherapy and for the development of new therapeutic approaches.

#### 2.2.1. Adenosinergic Pathway

Adenosine is an endogenous purine nucleoside present in most cells under physiological conditions due to its importance in several biological functions. Under stressful conditions, adenosine levels are increased, reducing tissue injury and promoting repair [48]. One of the major pathways of this extracellular increase during hypoxia is due to ATP release and subsequent cascade of ectonucleotidases, such as CD39 (nucleoside triphosphate diphosphohydrolase), which converts ATP to ADP/AMP and CD73 (5′-ectonucleotidase), hydrolyzing nucleotides into nucleosides and, thus, converting AMP to adenosine [49,50,51]. Adenosine is able to trigger its effects through four receptors, known as A1, A2A, A2B, and A3ARs. They belong to the superfamily of G-protein-coupled receptors (GPCRs), and they play an important role in the regulation of the intracellular production of cAMP. The G-protein subunit links the adenosine receptors to adenylyl cyclase. A1 and A3ARs are coupled to the inhibitory G-protein subunit, while A2A and A2BARs are coupled to the stimulatory subunit [49,52]. The role of A2AAR in cancerous tissues was described by Ohta et al. [53], where the levels of adenosine are high. They demonstrated that genetic deletion or the use of antagonists of A2AAR in immunogenic tumors achieved rejection or growth retardation in a CD8^+^ T-cell-dependent manner [53]. As hypoxia stimulates the production of adenosine, Hatfield et al. [54,55] showed that the use of hyperoxia blocks the hypoxia–HIF1α–[adenosine]high-A2AAR pathway, thus promoting tumor regression. Since CD8^+^ and CD4^+^ T cells avoid hypoxic areas, through this reversal, it is possible to prevent the inhibition of antitumor immunity, generating an immunopermissive TME with an enhancement of tumor infiltration by antitumor CD8^+^ but not CD4^+^ T cells [54,55,56].

Suppressive regulatory T cells (Tregs) are described to inhibit the antitumor immune response. They present a transcription factor known as FOXP3 necessary for their suppressive capacity and higher levels of CD39 [57]. The production of adenosine by Tregs is able to affect the functions of effector T cells, as well as their migration capacity to the tumor [58]. Moreover, the adenosine also has an effect over the Treg cells themselves; when hyperoxia is applied, there appears to be a reduction in their activity through the decrease in expression of CD39 and CD73, and consequently adenosine production. The levels of Tregs and Foxp3 expression are also decreased. The cytotoxic T-lymphocyte antigen-4 (CTLA-4) has an inhibitory role in the activity of the conventional T cells, and its levels in the Treg population significantly increase under the stimulation of A2AR, while supplemental oxygen is able to reduce its expression [55,59]. Hypoxia can also induce CD73 mRNA expression and functional protein levels, as well as CD39. After identifying the presence of a HIF-1α-binding site in the CD73 gene promoter, Synnestvedt et al. [50] demonstrated the role of HIF-1α in the induction of CD73, which is able to protect intestinal barrier during hypoxia by maintaining intestinal permeability [50]. These findings led to the development of a novel combined therapy against CD73 and HIF as an effective strategy using siRNA-loaded superparamagnetic iron oxide (SPION) nanocarriers [60]. During supplemental oxygenation, there is a reduction in the expression of these adenosine-generating ecto-enzymes, which could explain the decrease in adenosine levels [54].

These results support the idea of using therapies that combine the inhibition of HIF-1α and immunotherapies, in order to potentiate tumor rejection [56].

#### 2.2.2. MDSCs and PDL1

Myeloid-derived suppressor cells (MDSCs) are a subgroup of myelopoietic progenitor cells characterized by their immunosuppressive activity. In the context of tumor immune evasion, MDSCs play a critical role because of their ability to induce T-cell dysfunction, in addition to inhibiting B and NK cell activity [61]. The importance of MDSCs as key components of the TME has been shown through their expansion in multiple organs in a tumor-dependent and organ-specific way. Hypoxia is a leading characteristic of the TME, since it results in MDSCs suppressing antigen-specific and nonspecific T-cell activity via HIF-1α, although this does not occur in the peripheral lymphoid organs, where the MDSCs are only able to suppress the antigen-specific T-cell activity. The nonspecific suppression is due to the effect of HIF-1α on the upregulation of different proteins such as arginase and inducible nitric oxide synthase (iNOS), proteins that are recognized for their ability to suppress T-cell function. This factor also promotes MDSC differentiation to immune-suppressive TAMs (tumor-associated macrophages) [62]. 

The progression of colorectal cancer (CRC) is related to MDSCs. Despite it being known that these cells enhance the stemness of different cancer cells, the role that they have in CRC cell stemness is unclear. Exosomes produced during hypoxia in a HIF-1α-dependent manner by granulocytic MDSCs, which are the prevalent population of these myeloid cells, are able to stimulate CRC cell stemness. After oxygen treatment, there was not only a reduction in tumor growth in the mouse model, but also a decrease in HIF-1α levels, along with the number of exosomes secreted by these granulocytic cells [63]. There was an increase in the recruitment of MDSCs, as well as in the expression of their PD-L1 levels, in the premetastatic lung TME, generating an immunosuppressive microenvironment that was weakened through respiratory hyperoxia treatments. Although a decrease in PD-L1 expression was observed in hyperoxia-treated mice, MDSC recruitment was observed in the premetastatic phase but not metastatic phase [64]. Studying the deleterious consequences of hyperoxia on the lung, Hanidziar et al. [65] described their effects on the composition of immune cells in this organ by applying a CyTOF mass cytometry approach, highlighting their antitumor activity through the depletion of MDSCs from tumors. They also showed a depletion of regulatory B cells, which have an immunosuppressive effect, in part, due to the expression of high levels of PD-L1, consequently activating the PD-1 signaling pathway in target cells [65].

#### 2.2.3. PMNs

Polymorphonuclear neutrophils (PMNs) are the first line of defense against infection and the most abundant circulating immune cells [66]. Depending on the tumor microenvironment, they acquire different phenotypes that determine their role in cancer. It has been shown in a mouse model of endometrial cancer that hypoxia modulates the PMN phenotype. Specifically, it is able to recruit these cells to the tumor but prevent the development of their antitumoral functions, for instance, through the increase of tumor cell proliferation or limiting tumor cell death. The administration of respiratory hyperoxia improved their tumor control as reflected in a reduction in tumor burden, due to modifications in the interactions between tumor cells and PMNs, but not in a T-cell-dependent manner. Tumor oxygenation increases the production in the tumor epithelium of MMP-9- and NOX-2-derived ROS by PMNs compared with hypoxia. Both MMP-9 and NOX-2 are associated with basement membrane degradation and, therefore, trigger tumor cell death. Similarly, the relief from hypoxia seems to decrease cancer cell proliferation, reducing the effectiveness of neutrophil elastase, which has been previously described as a tumor proliferation stimulator [67,68]. Moreover, under this treatment, there is an increase in the population of PMNs with high transcript levels related to the functionality of MHCII complex, thus attributing antigen-presenting capacity to these cells. Due to the high heterogeneity of PMNs, the expression levels of SiglecF, a gene encoding the sialic acid-binding immunoglobulin-like lectin F, has been used to classify them into two subsets. A high expression of this marker was associated with tumor-promoting PMNs in lung cancer [69], and the use of hyperoxia treatment reduced its levels in a previously described uterus tumor model [68]. These results provide novel insights to reduce the hypoxia effects in cancer development using supplemental oxygenation.

## 3. Adverse Effects Derived from the Use of Hyperoxia

### 3.1. Hyperoxia-Induced Acute Lung Injury

The use of supplemental oxygen as a therapy is a lifesaving technique for seriously ill patients. The use of HBO treatment has been reported in different medical emergencies such as air embolism or CO poisoning, as well as recently in severe COVID-19 patients. Nevertheless, an prolonged exposition to hyperoxia could have detrimental effects, even leading to higher mortality in critically ill patients, especially those with extracorporeal life support and cardiac arrest [70,71]. It can trigger multiorgan injury, with the lung being the most affected, due to an intense inflammatory response and cell death that may result in acute lung injury (ALI). In addition, hyperoxia treatment in patients with myocardial injury promotes damage in the short term. However, clinical studies of the long-term effect on mortality have not clarified whether this occurs in a hyperoxic treatment-dependent manner [72,73]. As described above, when HBO is used, the oxygen pressure is higher, leading to an increase toxicity [37]. Despite the mechanisms of this event being poorly understood, it is accepted that ROS plays an essential role. Under physiological conditions, free radicals are scavenged by antioxidant enzymes; however, when hyperoxia occurs, they are not able to balance the free radicals due to the increase in intracellular level production. When prolonged hyperoxia takes place, not only are the levels of ROS increased, but also the NO production, due to an overactivation of NOS enzyme. Even if ROS production is necessary for the proper functioning of the cell, aberrant production could have a damaging effect on the mitochondria [74,75]. Studies performed in rat pulmonary epithelial cells (RLE-6TN) and rat lung tissues showed a decrease in the mitochondria membrane integrity through the loss of its potential, unleashing cell death. This change in the membrane potential has been demonstrated to increase apoptosis due to an imbalance between the proapoptotic protein Bax and antiapoptotic protein Bcl-2, since the loss of the membrane potential allows the liberation of cytochrome c from the membrane to the cytoplasm, which triggers the mitochondria-dependent apoptotic pathway. There is also an upregulation in inflammatory cytokines in an ROS/RNS-dependent manner, which recruit immune cells. Although there is still much to learn about this response, the NF-κB pathway and Toll-like receptors (TLRs) seem to be highly implicated. From the idea of ROS as an endogenous TLR ligand, Huang and colleagues showed for the first time in human alveolar epithelial tumor cells, A549, the role of the NF-κB pathway. This factor is known to be translocated into the nucleus as a result of the hyperoxia-induced oxidative stress response, in the inflammatory cascade. Inflammation is induced by intracellular ROS and consequently increasing IL-6 and IL-8 protein expression, as well as IFN-β- and IFN-dependent gene products. The use of scavengers against these species reverted the effects here described, thus confirming their role in the response. Furthermore, in vivo studies in a neonatal rat model confirmed the classic structure characteristic of lung injury [76,77].

In spite of the benefits of PMNs under hyperoxia in cancer treatment described above, this cell type is highly involved in this inflammatory response. High levels of this population are observed in a mouse model exhibiting lethal inflammatory responses to hyperoxia. Their migration is promoted by high ATP concentrations, as well as by different chemoattractants such as IFN-γ and IL-17 that are produced by iNKT cells (invariant natural killer T cells) after long supplemental oxygen exposure. These iNKT cells play an important role linking the innate and adaptive immune responses through their cytotoxic functions that are triggered after the recognition of lipids presented on CD1d by its invariant T-cell receptors [78,79]. They seem to increase their number in the lung and are stimulated by hyperoxia, producing IFN-γ and IL-17. The induction of lung injury could be due to CD39 activity, since this enzyme is highly expressed in iNKT cells, regulating their state of activation. This was demonstrated using CD39-null mice that showed protection during hyperoxia because the high concentrations of ATP triggered the activation of the ion channel P2X7, which caused iNKT cell death. In contrast, there was lung injury in the wildtype mice where active CD39 converted ATP into adenosine; therefore, the ligand did not bind to its receptor, supporting iNKT survival and proliferation [79]. 

There is no significant evidence of an ideal oxygen dose to treat different clinical conditions in ICU patients. Therefore, the mechanisms underlying the action of oxygen supplemental treatment need further investigation in order to establish an appropriate and beneficial treatment [80].

### 3.2. Protumor Inflammation

Inflammation encompasses a wide variety of physiological processes that enable the maintenance of homeostasis and protection against infection. On this matter, the correct functioning of the acute inflammation process is essential to reach an adequate recovery of the damaged tissue. However, when inflammation persists, it is involved in pathogeneses such as cardiovascular diseases or cancer. Thus, inflammation is considered a hallmark of cancer, named protumor inflammation, where its contribution to the hostile conditions of the tumor microenvironment induces tumor growth, invasion, and metastasis [81,82,83]. Although there are no studies demonstrating the relationship between hyperoxia treatment and protumor inflammation, given that supplemental oxygen s involved in inflammation, it could, in the long term, promote these mechanisms of tumor development and should, therefore, be taken into account in treatments involving prolonged exposure to oxygen.

Interleukin-1 (IL-1) is a proinflammatory cytokine with a high relevance in cancer; 11 ligands and receptors have been described in this family, where IL-1β has high relevance in the inflammation process [84]. In a lung metastasis model using lung adenocarcinoma A549 cells, this molecule enhanced the expression of adhesion molecules that facilitate the binding of cancer cells to endothelial cells, as well as metalloproteinases, capable of degrading the extracellular matrix of basement membranes. A high IL-1 mRNA expression was detected in different cancer cell lines, seemingly related to increased endothelial permeability. The importance of this molecule in tumor growth and metastasis is reflected in a reduction in these parameters when an antagonist is used. Specifically, in the production of IL-1β, the innate immune pathway through the NLRP3 (NOD-like receptor protein 3) inflammasome plays a critical role. The components of this complex were highly expressed in head and neck squamous cell carcinoma (HNSCC) tissue samples, as well as in a mouse model. In this model, when a small-molecule inhibitor was used, the production of IL-1β was reduced with tumor growth. Inflammasome inhibition also modifies the TME toward an antitumoral one with a decrease in immunosuppressive cells and an increase in effector T cells [85]. 

Stromal cells and particularly fibroblasts have great importance in shaping the TME. The special cells known as cancer-associated fibroblasts (CAFs) are able to suppress T cells in different cancer models [86,87]. The importance of IL-1β in cancer invasion was shown in a squamous cell carcinoma model through the capacity of inducing TNF-α expression. TNF-α produced by cancer cells is a cue to improve the ability of CAFs to model the extracellular matrix (ECM) [88]. CAFs induced protumor macrophages, as well as the suppression of T cells in an oral squamous cell carcinoma model. Among all the cells present in the tumor microenvironment, the more abundant immune cells are the macrophages or TAMs. In patients with this type of cancer, a high number of CAFs correlated with a high presence of TAMs. Furthermore, these patients expressed more clinicopathological features such as vascular invasion, lymphatic invasion, or metastasis when they presented higher levels of CAFs [89,90]. 

IL-1β is able to upregulate the accumulation of MDSCs, although these cells lack the IL-1R. This suggests that this proinflammatory environment promotes necessary pathways that end in tumor progression facilitated by MDSCs, as observed in the higher tumor growth and survival rate of an IL-1β-injected mammary carcinoma mice model [91]. The effect of the transgenic expression of the cytokine in the stomach of a mouse model resulted in a spontaneous carcinoma, in line with the recruitment of MDSCs to the site. However, in this model, the expression of the IL-1R was proven in MDSCs. These results suggest a direct effect of the IL-1β on the activation of MDSCs, resulting in an activation of the NF-κB pathway [92].

In addition, inflammation has been related to pathological, but not physiological angiogenesis, and both processes have been associated with multiple alterations such as cardiovascular diseases and cancer. VEGF-A, the main regulator of tumor angiogenesis, is upregulated and produced by malignant cells under hypoxic and/or inflammatory conditions [93,94,95,96]. The effects of the VEGF family are mediated by their binding to two kinds of VEGF receptors (VEGFR): VEGFR1 and VEGFR2. VEGFR1 is present in endothelial cells and in multiple kinds of myeloid populations such as immature myeloid cells (iMCs) and macrophages. VGFR1 signaling in myeloid cells has been implicated in cell migration to tumors and inflammatory niches, promoting tumor progression and angiogenesis. However, VEGFR1 works as a VEGF decoy receptor in endothelial cells. VEGFR2 is present in endothelial cells and is implicated in their proliferation and migration processes [96,97,98]. IL-1β was also described to be required for in vivo invasiveness and angiogenesis in several tumor cell lines [99,100]. Carmi and colleagues discovered the relationship between IL-1β and VEGF in early angiogenesis during tumorigenesis, confirming how inflammation and tumor angiogenesis work together. They described that the recruitment of myeloid cells such as neutrophils, macrophages, and MDSCs to the tumor microenvironment by tumor cells promotes the secretion of active IL-1β and other proinflammatory mediators such as CCL2, CCL3, and Bv8. This process in turn induces the activation and secretion of VEGF and other proangiogenic molecules such as PIGF, PDGF, and bFGF by the tissue-resident endothelial cells. Therefore, there is a crosstalk between proinflammatory mediators produced mainly by VEGFR1^+^ iMCs and proangiogenic factors secreted by VEGFR2^+^ endothelial cells, which complement each other’s function during the early angiogenic response [101].

This cytokine was also shown to promote angiogenesis in breast cancer through the induction of HIF-1α, which promoted the transcription of VEGF [102]. Another study focusing on the clinical implications of serum VEGF-A and IL-1 β in patients with advanced non-small-cell lung cancer (NSCLC) revealed that high levels of those cytokines were related to shorter overall survival, revealing them as significant prognostic factors [103]. Both angiogenesis and inflammation are also linked because many proangiogenic proteins present proinflammatory properties, and many proinflammatory factors have proangiogenic effects. Whereas the contribution of inflammatory cells is well known, the role of endothelial cells is less understood. For this reason, a proteomic study comparing quiescent, angiogenic, and inflammatory activated endothelial cells was performed. HUVECs were stimulated with VEGF and IL-1β, and the secreted, cytoplasmic, and nuclear fractions were obtained and processed to obtain proteomic profiles. This study showed in detail the activation process for each cellular compartment, becoming a reference proteome which identified new potential genes and providing a computational data framework for future proteomic studies [83].

In a hypoxic tumor, the formation of an immune-permissive niche may occur in an aberrant vasculogenesis due to the overproduction of VEGF, which is able to potentiate the activity of several types of immune cells such as TAMs or MDSCs. In accordance with the strong crosstalk between angiogenesis and the immune system that we previously described, there is evidence showing the great efficacy in overcoming the resistance to antiangiogenic therapies when using a combinatory therapy against angiogenic and immune processes [104,105].

Taking all this into consideration, the development of combined therapies to keep inflammation under control in clinical settings using hyperoxia seems highly necessary.

## 4. Transcriptional Profile

We performed RNA sequencing (RNA-seq), following the Material and Methods described in Appendix A, to analyze the effect of hyperoxia treatment on the transcriptional profile of tumor cells compared to healthy cells. For this purpose, total RNA extraction was performed using two types of cells: MUM2B metastatic uveal melanoma cells and HBEC3-KT cells (normal human bronchial epithelial cells). These cells were cultured under normoxic or hyperoxic conditions. Hyperoxic incubation under 30% O_2_ was achieved using a ProOx C21 Oxygen and Carbon Dioxide Subchamber Controller (BioSpherix, Ltd., New York, NY, USA, RRID: SCR_021131) [106,107,108,109,110,111,112,113,114,115].

When comparing RNA transcripts of MUM2B cells, we observed a downregulation in the expression of 1267 genes and upregulation of 1495 genes when hyperoxia was applied, involving 2762 genes globally. The enrichment analysis showed that the main affected genes are involved in mitochondrial or translational biological processes (Figure 3). The volcano plot presented in Figure 4a shows statistically significant genes (Appendix A). Under hyperoxia, a reduction in the expression of genes that encode mitochondrial electron transport chain-associated proteins was observed, including different subunits of Complex I, such as those encoded by MT-ND4L, MT-ND1, NADH-ubiquinone oxidoreductase chain 4L, or NADH-ubiquinone oxidoreductase chain 1, as well as of Complex IV, such as the cytochrome c oxidase subunit 3 encoded by MT-CO3. 

In contrast, we highlight the overexpression of genes related to cell adhesion, such as CLDN16 or ITGB4, encoding Claudin-16 and Integrin beta-4, respectively. The latter is crucial for hemidesmosome formation; different members of the keratin gene family, such as Keratin 5 or Keratin 17 (genes KRT5 or KRT17), are associated with cell growth and differentiation. We also found the upregulation of genes involved in the regulation of the transcription, such as the component of the nucleosome known as Histone H2A type 1, encoded by H2AC11.

These results confirm that high oxygen levels or exposure have a detrimental effect on the mitochondria.

Nevertheless, when this comparison was performed in HBEC3-KT cells, we saw a huge decrease in the number of changes in gene expression. The total number of genes was 109, whereby 37 of them diminished their expression while 72 were highly expressed.

The most significant genes are presented in the volcano plot (Figure 4b and Appendix A). Among those downregulated genes, we can find genes involved in transcription processes as ZNF174, encoding zinc finger protein 174, which is a repressor. There was also a reduction in the expression of TANK, encoding the TRAF family member-associated NF-kappa-B activator, which is a negative regulator of NF-κB signaling. 

The analysis of the upregulated genes revealed that the expression of genes involved in protein processing such as SEC31B, encoding the protein transport protein Sec31B, and the LIM/homeobox protein Lhx4, encoded by the LHX4 gene, is crucial for the development of respiratory mechanisms and correct lung development.

However, it was not possible to predict the biological processes modified due to the low number of affected genes, which did not allow developing significantly altered pathways.

These results could be explained by the higher tolerance of the lung cells to oxygen, as no pathway was altered under this treatment, whereas the malignant cells needed to modify their metabolism in order to maintain homeostasis.

Although further research is needed, these results indicate that the use of hyperoxia treatment to counteract tumor progression would have a greater effect on malignant cells, opening up therapeutic opportunities that need to be further explored.

## 5. Conclusions

Several studies have demonstrated the benefits of supplemental oxygen treatment to avoid tumor progression. Both chemotherapy and radiotherapy present a synergic effect with hyperoxia. In addition, several cellular pathways are modified under this condition, resulting in the modification of the immune response toward an immunopermissive TME. Of particular interest are the effects of hyperoxia on the immune system and the therapeutic opportunities for improving current immunotherapy-based approaches. Of particular note is the adenosinergic pathway, in which the use of adenosine receptor scavengers or inhibitors, as well as the use of supplemental oxygen, achieves tumor regression in animal models. 

However, prolonged exposure appears to produce an excessive inflammatory response that could result in organ injury, where the most affected may be the lung. Further research is needed to develop new combined therapies targeting the altered immune pathways under supplemental oxygen treatment in order to achieve more targeted therapies against tumor cells to prevent the detrimental effects of this therapy.

In a global analysis of the transcriptional profile of human uveal melanoma cells, which usually metastasize to the lung, compared to healthy lung epithelial cells, we observed that hyperoxia affects healthy cells much less than malignant cells. Tumor cells are more profoundly affected in mitochondrial functions, which opens new possibilities for exploration, to analyze to what extent their cellular metabolism is affected and whether it is possible to take advantage of this circumstance for the development of new therapies.

## Figures and Tables

**Figure 1 cancers-14-02740-f001:**
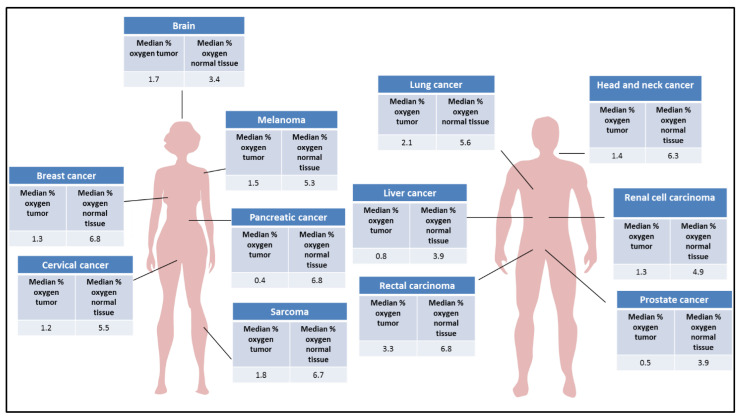
Median percentage oxygen in human tumors and related normal tissues.

**Figure 2 cancers-14-02740-f002:**
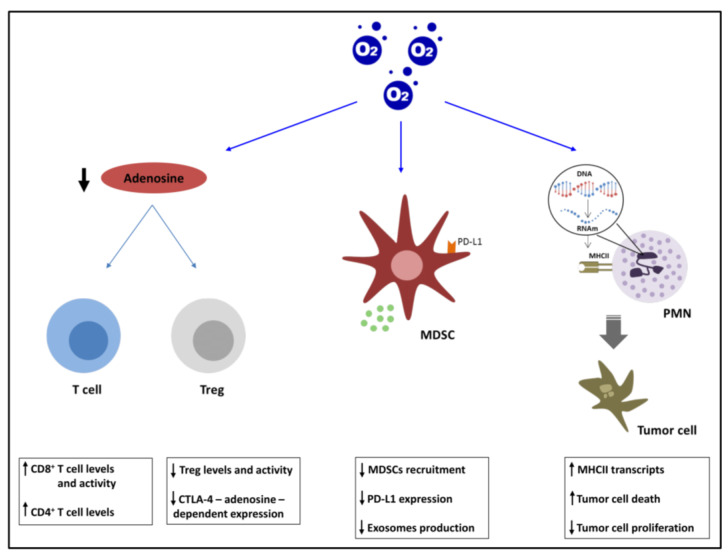
Hyperoxia treatment effect over the immune response. The adenosinergic pathway is altered, presenting a modification in the activity of the effector and regulatory T cells. MDSC recruitment and activity are modified, while PMNs are altered, leading to an immunopermissive TME.

**Figure 3 cancers-14-02740-f003:**
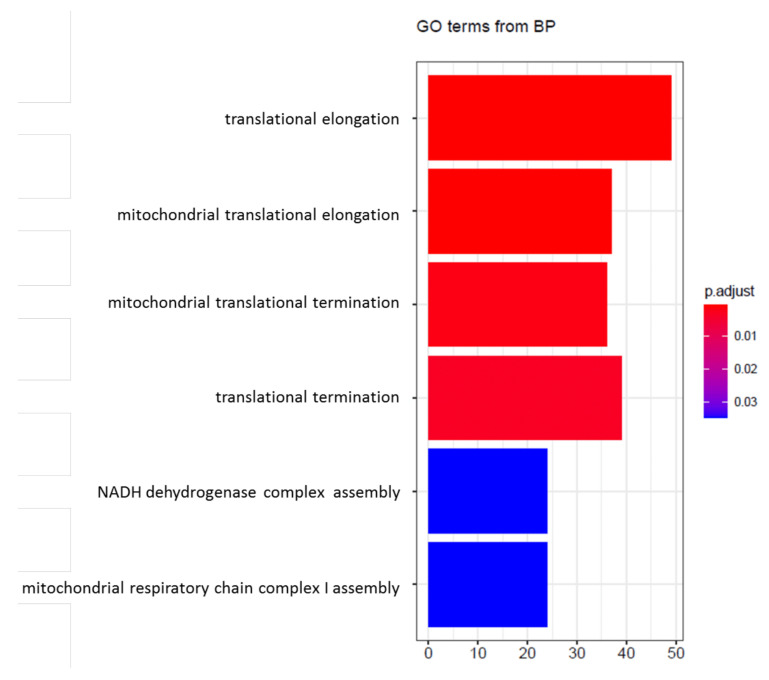
GO terms from biological processes found to be mainly modified following the gene enrichment analysis.

**Figure 4 cancers-14-02740-f004:**
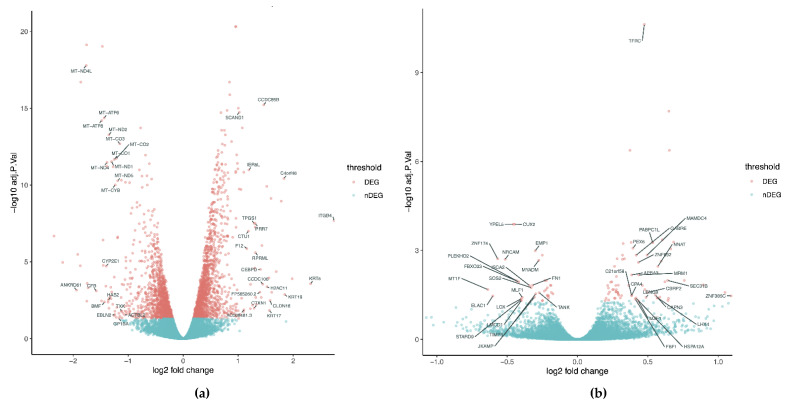
Volcano plot representation of the main genes whose expression significantly changed under hyperoxic conditions in (**a**) uveal melanoma cell line (MUM2B) and (**b**) normal human bronchial epithelial cells (HBEC3-KT) (Appendix A).

## Data Availability

The RNA-Seq data discussed in this publication are accessible through SRA accession number PRJNA825377.

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
