# Peer review of "Implications of Hyperoxia over the Tumor Microenvironment: An Overview Highlighting the Importance of the Immune System"

_cancers, 2022, doi:10.3390/cancers14112740_

Round 1

Reviewer 1 Report

The review paper described a special topic about hyperoxia on tumor progression, according to their title and text contents. This is helpful for the readers to understand that hypoxia is crucial for tumor microenvironment and related to metastasis, resistance and clinical prognosis. At the same time, the authors summarized how hyperoxia could be helpful to prevent tumor progression, including the impact of hyperoxia on different cancer therapies, adverse effects of hyperoxia, and transcriptional profile of tumor cells after hyperoxia treatment. Especially, the transcriptional profile will guide the future studies and possibly make this method more practical in clinical applications. 

There are several suggestions to the authors about the manuscript:

  1. More references should be included in this manuscript. For examples, (1) Dong D, Fu Y, Chen F, et al. Hyperoxia sensitizes hypoxic HeLa cells to ionizing radiation by downregulating HIF‑1α and VEGF expression. Mol Med Rep. 2021;23(1):62. doi:10.3892/mmr.2020.11700; (2) Kim SW, Kim IK, Ha JH, et al. Normobaric hyperoxia inhibits the progression of lung cancer by inducing apoptosis. Exp Biol Med (Maywood). 2018;243(9):739-748. doi:10.1177/1535370218774737; (3) Ristescu AI, Tiron CE, Tiron A, Grigoras I. Exploring Hyperoxia Effects in Cancer-From Perioperative Clinical Data to Potential Molecular Mechanisms. Biomedicines. 2021;9(9):1213. Published 2021 Sep 13. doi:10.3390/biomedicines9091213; (4) Kim SW, Kim IK, Lee SH. Role of hyperoxic treatment in cancer. Exp Biol Med (Maywood). 2020;245(10):851-860. doi:10.1177/1535370220921547; (5) Singer et al. Critical Care (2021) 25:440. https://doi.org/10.1186/s13054-021-03815-y. These papers should be properly cited in current manuscript.
  2. Several typos need corrections. (1) Subtitle "2.1.1. Radiotherapy" should be "2.1.2. Radiotherapy". (2) Pay attention to subscripts, O2, "2" should be subscript. (3) In part 4 of the manuscript, "We performed RNA sequencing....", the authors used "we" here will lead to some confusions, these data were not from the authors of current manuscipt, but from the references [95-104].
  3. Are there any reports or references about the recovery of the transcriptional profiles after the hyperoxia? One week later or more longer times after hyperoxia treatment, will the altered genes returned to normal level?

Reviewer 2 Report

The goal of this review is to provide an overview of how hyperoxia interacts with other therapies decreasing tumor progression and the negative effects of the use of high oxygen levels. They also perform an analysis, showing the differences in the patterns of expression between a tumor‐derived cell line and a non‐malignant cell line.

This review is a limited summary of this topic and was preceded by several reviews which are not cited (see attached list). The article is a mixed article, half review and half experimental results. It is recommended that the experimental results will be published separately.

The review should focus on chapter 2.2 “cancer immunotherapy and hyperoxia” which should become “cancer immunology and hyperoxia. This will also explain the need for Chapter 3.2. “Pro‐tumor Inflammation” which at the present form seems to be out of context and is a complicated paragraph.

The focus on tumor immunology and TME should also be reflected in the title.

References

Effects of hypoxia and hyperoxia on the differential expression of VEGF-A isoforms and receptors in Idiopathic Pulmonary Fibrosis (IPF).

Barratt SL, Blythe T, Ourradi K, Jarrett C, Welsh GI, Bates DO, Millar AB.Respir Res. 2018 Jan 15;19(1):9. doi: 10.1186/s12931-017-0711-x.PMID: 29334947 Free PMC article.

Potential of hyperbaric oxygen in urological diseases.

Tanaka T, Minami A, Uchida J, Nakatani T.Int J Urol. 2019 Sep;26(9):860-867. doi: 10.1111/iju.14015. Epub 2019 May 13.PMID: 31083787 Review.

Role of hyperoxic treatment in cancer.

Kim SW, Kim IK, Lee SH.Exp Biol Med (Maywood). 2020 May;245(10):851-860. doi: 10.1177/1535370220921547. Epub 2020 Apr 23.PMID: 32326758 Free PMC article. Review.

Antihypoxic oxygenation agents with respiratory hyperoxia to improve cancer immunotherapy.

Hatfield SM, Sitkovsky MV.J Clin Invest. 2020 Nov 2;130(11):5629-5637. doi: 10.1172/JCI137554.PMID: 32870821 Free PMC article. Review.

A General Overview on the Hyperbaric Oxygen Therapy: Applications, Mechanisms and Translational Opportunities.

Ortega MA, Fraile-Martinez O, García-Montero C, Callejón-Peláez E, Sáez MA, Álvarez-Mon MA, García-Honduvilla N, Monserrat J, Álvarez-Mon M, Bujan J, Canals ML.Medicina

(Kaunas). 2021 Aug 24;57(9):864. doi: 10.3390/medicina57090864.PMID: 34577787 Free PMC article. Review.

Exploring Hyperoxia Effects in Cancer-From Perioperative Clinical Data to Potential Molecular Mechanisms.

Ristescu AI, Tiron CE, Tiron A, Grigoras I.Biomedicines. 2021 Sep 13;9(9):1213. doi: 10.3390/biomedicines9091213.PMID: 34572400 Free PMC article. Review.

Hyperbaric oxygen therapy for malignancy: a review. Daruwalla J, Christophi C.World J Surg. 2006 Dec;30(12):2112-31. doi: 10.1007/s00268-006-0190-6.PMID: 17102915 Review.

Reviewer 3 Report

The topic reviewed is of interest, nonetheless, the following points need to be considered:

1) The rationale of why the authors came up with this review.

2) What is the information that is not exactly available that motivated the authors to come up with this information. What are the current caveats and how do the authors highlight the current research in answering them? If not they need to address it in future directions.

3)As is now well known, tumors grow and evolve through a constant crosstalk with the surrounding microenvironment, and emerging evidence indicates that angiogenesis and immunosuppression frequently occur simultaneously in response to this crosstalk: please expand.

4) In the frame of point 3 thinking, strategies combining anti-angiogenic therapy and immunotherapy seem to have the potential to tip the balance of the tumor microenvironment and improve treatment response. (please refer to PMID: 32456352 and expand).

5. Moreover, e association of multiple anti-angiogenic molecules or a combination of anti-angiogenic drugs with other treatment regimens have been indicated as alternative therapeutic strategies to overcome resistance to anti-angiogenic therapies. Alternative mechanisms of tumor vasculature, including intussusceptive microvascular growth (IMG), vasculogenic mimicry, and vascular co-option, are involved in resistance to anti-angiogenic therapies. The crosstalk between angiogenesis and immune cells explains the efficacy of combining anti-angiogenic drugs with immune check-point inhibitors. Collectively, in order to increase clinical benefits and overcome resistance to anti-angiogenesis therapies, pan-omics profiling is key (please refer PMID: 34298648).6. A workflow scheme or a graphical abstract might improve the manuscript quality.

Round 2

Reviewer 2 Report

I have no further comments for the authors

Reviewer 3 Report

The authors have clarified several of the questions I raised in my previous review. Most of the major problems have been addressed by this revision.